# Creep Feeding and Weaning Influence the Postnatal Evolution of the Plasma Metabolome in Neonatal Piglets

**DOI:** 10.3390/metabo13020214

**Published:** 2023-01-31

**Authors:** Barbara U. Metzler-Zebeli, Frederike Lerch, Fitra Yosi, Julia C. Vötterl, Simone Koger, Markus Aigensberger, Patrick M. Rennhofer, Franz Berthiller, Heidi E. Schwartz-Zimmermann

**Affiliations:** 1Unit of Nutritional Physiology, Department of Biomedical Sciences, University of Veterinary Medicine Vienna, 1210 Vienna, Austria; 2Christian-Doppler Laboratory for Innovative Gut Health Concepts of Livestock, University of Veterinary Medicine Vienna, 1210 Vienna, Austria; 3Department of Animal Science, Faculty of Agriculture, University of Sriwijaya, Palembang 30662, South Sumatra, Indonesia; 4Department for Farm Animals and Veterinary Public Health, Institute of Animal Nutrition and Functional Plant Compounds, University of Veterinary Medicine Vienna, 1210 Vienna, Austria; 5Department of Agrobiotechnology (IFA-Tulln), Institute of Bioanalytics and Agro-Metabolomics, University of Natural Resources and Life Sciences, Vienna (BOKU), 3430 Tulln an der Donau, Austria

**Keywords:** plasma metabolome, neonatal piglets, triglycerides, hippuric acid, creep feeding, weaning

## Abstract

Data on the evolution of blood metabolites and metabolic markers in neonatal piglets are scarce, although this information is vital to detect physiological aberrations from normal development. We aimed to characterize age- and nutrition-related changes in the plasma metabolome and serum biochemistry of suckling and newly weaned piglets and assess metabolite patterns as physiological markers for the two phases. In two replicate batches (*n* = 10 litters/group), piglets either received sow milk alone or were additionally offered creep feed from day 10 until weaning (day 28). Blood was collected from one piglet/litter on days 7, 14, 21, 28, 31 and 35 of life, totaling five females and five males/group/day. Signature feature ranking identified plasma triglycerides (TG) as discriminative for age and nutrition during the suckling phase. Influential TG 20:4_36:5, TG 17:0_34:2 and TG 18:2_38:6 were higher in creep-fed piglets on days 14, 21 and 28 of life, respectively, compared to only sow milk-fed piglets. Metabolites belonging to pathways within histidine, D-glutamine and D-glutamate metabolism as well as hippuric acid were distinctive for the postweaning compared to the suckling period. In conclusion, plasma lipid profiles especially corresponded to the type of nutrition in the suckling phase and showed a strong weaning effect.

## 1. Introduction

Metabolomics approaches have been used to detect markers for a variety of health and physiological conditions in animals [1,2]. While there is an extensive body of research on weaned and growing pigs [3,4], the developmental patterns in the blood metabolome of neonatal piglets from the suckling to early postweaning phase have been incompletely described. Knowledge about the ranges of metabolites and metabolic markers used in routine diagnostics (e.g., total protein, alanine transaminase (ALT), aspartate transaminase (AST) and alkaline phosphatase (ALP)) at this age can provide valid data about the physiological state and health of piglets. Metabolite ranges can be different in infants and adults, requiring the establishment of appropriate reference ranges [5]. Therefore, to understand alterations in the metabolic phenotype [6], knowledge on the normal changes in the plasma metabolome in different life phases is fundamental.

Piglets from modern hyperprolific breeds more than quadruple their birth weight during the 28-day-long suckling phase. This growth spurt demands a high turnover of nutrients. Concurrently, the energy metabolism shifts from being glucogenic to becoming ketogenic postpartum [7]. Late colostrum and transient milk are especially rich in fat [8], which is matched by the secretion of potent lingual, gastric and pancreatic lipases by the piglet, ensuring maximum digestive utilization of the fat [9]. After weaning, the piglet is transferred to a high-starch, low-fat diet, shifting the metabolism back to glucogenic mode and respective metabolites. Moreover, the low feed intake in the first days after weaning and the upregulation of the hypothalamic–pituitary–adrenal stress response increase catabolic processes and mobilization of body tissue [10], which may be noticeable in the plasma metabolome. However, conclusive metabolome profiles for the suckling and early postweaning phase in pigs are missing.

Aside from sow milk, piglets are fed highly palatable and digestible creep feed as early as day 2 of life to ease the transition from liquid to solid feed after weaning. In addition, creep feeding can ensure adequate nutrition when sow milk production cannot adequately meet the needs of her litter which has become more important due to big litter sizes [11]. The chemical composition of creep feed differs from porcine milk, containing a considerable amount of starch and less fat than porcine milk which can be attributed to ingredients like cow dairy by-products, such as whey powder and lactose, plant protein (e.g., soy protein hydrolysate), starches and plant oils. Consequently, if the consumption of creep feed is high enough, creep feed may alter the profiles of metabolites and metabolic markers in blood, which has not been sufficiently described before but may need to be considered especially in relation to routine diagnostics in veterinary practice. Piglets are changed to the cereal-based prestarter diet in the late suckling phase, which is then fed until after weaning, again modulating the intestinally and metabolically available nutrient composition. However, little information is available on whether blood metabolite profiles, when determined during the suckling and early postweaning period, are indicative for the nutrition of the piglets preweaning.

We hypothesized that the introduction of creep feed would modulate the age-related metabolome profile, especially influencing the utilization of lipids as energetic substrate. Assessing this hypothesis could provide insights into age-related changes in the associated biochemical processes, and possibly reveal physiological biomarkers for the nutritional state of piglets. We further hypothesized that blood metabolite patterns in the early days postweaning would shift dramatically due to the removal of porcine milk and lower feed intake in the first days after weaning but still would allow distinguishing which piglets received creep feed. Therefore, the objective of the present study was to characterize age- and nutrition-related changes in the plasma metabolome as well as in additional parameters for clinical chemistry in the serum of suckling and newly weaned piglets.

## 2. Materials and Methods

### 2.1. Animals, Housing and Feeding

The animal experiment was conducted under practical conditions at the pig facility of the research and teaching farm ‘VetFarm’ of the University of Veterinary Medicine Vienna (Vienna, Austria) in two replicate batches, with 10 sows per replicate batch. The sow handling and feeding were similar in both replicate batches. The sows, as part of the University sow herd, were fed and handled similarly during gestation and lactation according to standard procedures at the farm. Five days before farrowing, sows were moved into free-ranging farrowing pens (BeFree, Schauer, Agrotonic, Prambachkirchen, Austria; 2.3 × 2.6 m) including a bowl drinker, feeder and hayrack for the sow and bowl drinkers and a nest with heated flooring for the piglets in one farrowing room. Sows were not constrained for farrowing. The farrowing pens comprised a nest for the piglets with heated flooring. Cross-fostering was minimized and applied only on day 1 of life to keep litter size at a maximum of 13 piglets. Mainly small birth weight piglets were cross-fostered to sows that were not included in the experiment. The other piglets remained with their mother sows throughout lactation.

Piglets were individually weighed and identified with an ear tag after birth. Piglets received an iron injection on day 4 of life (2 mL Ferriphor 100 mg/mL, OGRIS Pharma Vetriebs-GmbH, Wels, Austria) and vaccination on day 17 of life (1 mL Ingelvac Circoflex plus 1 mL Ingelvac MycoFLEX, Boehringer Ingelheim RCV GmbH und Co KG, Vienna, Austria). Male piglets were castrated on day 11 of life after sedation (Stresnil 40 mg/mL, 0.025 mL/kg body weight, Elanco Tiergesundheit AG, Basel, Switzerland and Narketan 100 mg/mL, 0.1 mL/kg body weight, Vetoquinol Österreich GmbH, Vienna, Austria). On day 28 of life, the sows were first removed from the farrowing room, before piglets were transferred to the weaner pig room. Piglets were kept in groups of a maximum of 20 piglets (pen size 3.3 m × 4.6 m), whereby piglets from two to three litters of the same dietary group were housed together. Pens were equipped with a piglet nest, nipple and bowl drinkers and one round feeder. Straw was provided as bedding material. Sows and piglets had free access to water.

Throughout lactation, sows were fed a commercial cereal–soybean meal-based lactation diet (Appendix A). Piglets from all litters suckled freely during the 28-day suckling period. The experimental outline is shown in Figure 1. The piglets from five litters per replicate batch suckled only sow milk (sow milk group). The piglets from the other five litters per replicate batch additionally had free access to a commercial milk replacer as creep feed from day 10 of life (creep feed group). The dry powder was mixed 1:5 (wt/vol; 200 g/L) with warm water (45 °C) to achieve a thin liquid according to the manufacturer’s instructions that was manually served and offered in piglet feed troughs at least twice daily (0800 and 1500 h). On days 24 and 25 of life the creep feed was gradually mixed with the prestarter diet (Appendix A), after which the piglets from the creep feed group were fed 100% of the prestarter diet. After weaning on day 28 of life, the prestarter diet was fed to 100% until day 35 of life, which marked the end of the experiment. Litters in the sow milk group (n = 5/replicate batch) did not receive the prestarter diet before weaning and were offered the prestarter diet from day 28 to 35 of life. Piglets in both feeding groups had free access to the prestarter diet postweaning. All diets were commercial complete feeds and met the current recommendations for nutrient requirements [12].

### 2.2. Body Weight Measurements and Blood Sampling

Body weight (BW) of the piglets was measured on days 1, 3, 6, 13, 20, 27, 30 and 34 of life. In each of the two replicate batches, blood samples were collected from ten piglets each at day of life 7, 14, 21, 31 and 35. On each of the five sampling days, five piglets per diet and sex were selected based on average body weight development across litters. Blood was taken from the heart on days 7, 14, 21, 28, 31 and 35 of life after deep sedation of the piglets (Stresnil 40 mg/mL, 0.025 mL/kg body weight, Azaperone, Elanco Tiergesundheit AG, Basel, Switzerland and Narketan 100 mg/mL, 0.1 mL/kg body weight, Ketamine, Vetoquinol Österreich GmbH, Vienna, Austria). The blood was collected into plasma and serum tubes (Vacuette Röhrchen K3E K3EDTA and Vacuette Röhrchen CAT Serum, Greiner Bio-One International GmbH, Kremsmünster, Austria) for metabolomics and clinical biochemistry, respectively. After inverting the blood tubes, they were stored on crash ice until centrifugation at 3000× *g* for 20 min at 4 °C (Eppendorf Centrifuge 5810 R, Eppendorf, Hamburg, Germany). Serum and plasma were aliquoted and stored at −80 °C until analysis.

### 2.3. Metabolomics

Solvents and reagents for metabolomics analysis were of analytical grade or higher and obtained from VWR International GmbH (Vienna, Austria), Sigma-Aldrich (Vienna, Austria), and Merck (Darmstadt, Germany). Water was purified with an Arium^®^ pro Ultrapure Lab Water System (Sartorius, Göttingen, Germany). Reference standards for all analytes were purchased from Sigma-Aldrich (Vienna, Austria), VWR (Vienna, Austria), Cayman Chemicals (Ann Arbor, MI, USA) and Avanti Polar Lipids (Birmingham, AL, USA, products available through Merck KGaA, Darmstadt, Germany). ^13^C-labelled internal standards were obtained from Sigma-Aldrich, deuterated bile acids from Sigma-Aldrich and Avanti Polar Lipids, and a deuterated cell-free amino acid mix containing 20 amino acids at different ratios was purchased from Eurisotop (Tewksbury, MA, USA).

Metabolites in plasma were quantitatively determined by three reversed phase high-performance liquid chromatography tandem mass spectrometric (RP-HPLC-MS/MS) methods and one anion exchange chromatography high-resolution mass spectrometric (AIC-HR-MS) method. Amino acids, biogenic amines and most lipid classes were measured by tandem mass spectrometry in positive ionization mode after RP-HPLC separation, whereas bile acids as well as medium and long chain fatty acids were determined by RP-HPLC-MS/MS in negative ionization mode. Carboxylic acids, sugar phosphates and nucleotides were measured by AIC-HR-MS. An overview of the investigated compounds and the analyses methods used for their determination is given in Appendix A.

For all compounds measured by RP-HPLC-MS/MS, sample preparation was performed in 96-well plates using a modified protocol based on Biocrates’ MxP^®^ Quant 500 kit (Biocrates, Innsbruck, Austria) that includes derivatization of amino acids and amines with phenyl isothiocyanate (PITC). Aliquots of 10 µL of plasma or different volumes of calibration solutions containing between 0.009 and 16 mg/L of analytes and 25 µL of internal standard solution (8.3 mg/L of ^13^C-putrescine, 740 mg/L of labelled amino acid stock solution, 0.5–2.0 mg/L of deuterated bile acids) were pipetted into a 96-well plate and evaporated to dryness in a Centrivap vacuum concentrator (Labconco, Kansas City, MO, USA) at 30 °C. Afterwards, 50 µL of derivatization reagent (ethanol/water/pyridine/PITC 31.7/31.7/31.7/5.0, *v*/*v*/*v*/*v*) was added and the covered plate was shaken for 20 s and placed in the dark at ambient temperature for derivatization of amino acids and amines. After 1 h, the derivatization reagent was evaporated in the Centrivap concentrator. Finally, analytes were extracted into 300 µL of methanol containing 4.9 mM ammonium acetate by shaking for 30 min and the extracts were centrifuged. One aliquot of the extracts was used directly for LC-MS/MS measurement in the negative ionization mode, while another aliquot was diluted 1 + 4 (v + v) with methanol for determination of amino acids, amines and lipids in the positive ionization mode.

The RP-HPLC-MS/MS analysis was performed on an Agilent 1290 UHPLC system (Agilent Technologies, Waldbronn, Germany) coupled to a QTrap 6500+ mass spectrometer equipped with an IonDrive TurboV source (SCIEX, Foster City, CA, USA). Chromatographic separation was carried out on a Kinetex C18 column (50 × 2.1 mm, 1.7 μm particle size, Phenomenex, Aschaffenburg, Germany) at a temperature of 50 °C. Three different chromatographic and mass spectrometric methods were applied for analysis of amino acids and amines (scheduled selected reaction monitoring mode, sSRM, positive ionization mode), bile acids and fatty acids (SRM, negative ionization mode) and lipids (SRM, positive ionization). The chromatographic methods are summarized in Appendix A. The ion source parameters for MS detection in positive mode were as follows: source temperature 500 °C, ion spray voltage 5500 V, curtain gas 45 psi, ion source gas 1 60 psi and ion source gas 2 70 psi. In negative mode, the ion spray voltage was reduced to −4500 V. SRM transitions are given in Appendix A. Analyst software version 1.6.3 (SCIEX, Framingham, MA, USA) was employed for instrument control and data analysis. Quantitative analysis was performed on the basis of linear or quadratic calibration curves in the range between 0.6 and 3000 ng/mL of derivatized reference standard compounds in measurement solution. Internal standards were used for recovery correction.

In addition to quantitative determination of compounds with a reference standard, semi-quantitative analysis of several lipid classes without an available reference standard (acyl carnitines, phosphocholines, ceramides, cholesterol esters, sphingomyelins, diglycerides and triglycerides) was also performed. To that end, the LC method used for quantification of lipids with a reference standard was paired with SRM transitions of various lipid classes without a reference standard. The transitions used were the same as in the flow injection part of Biocrates’ MxP^®^ Quant 500 kit (Biocrates, Innsbruck, Austria). Semi-quantitative analysis was achieved on the basis of molar calibration curves for compounds of the same lipid class with available reference standard.

Sample preparation for AIC-HR-MS consisted of shaking 20 µL of plasma with 10 µL of AIC internal standard solution (5 mg/L of fully ^13^C-labelled acetic acid, propionic acid and butyric acid) and 470 µL of acetonitrile/water (80/20, *v*/*v*) at 4 °C for 10 min and centrifugation at 14,350× *g* for 10 min. The AIC-HR-MS measurements were carried out on a Dionex Integrion HPIC system coupled to a Q Exactive Orbitrap mass spectrometer (both Thermo Scientific, Waltham, MA, USA) as described in our previous work [13]. Analytes were quantified on the basis of pure solvent calibration curves established between 0.1 and 9000 ng/mL for all analytes. Hexoses and disaccharides that were co-eluting under the chosen conditions were quantified as sum parameters using glucose and sucrose, respectively, as reference standard.

### 2.4. Clinical Biochemistry

The serum content of calcium, potassium, sodium, chloride, phosphate, total protein, ALT, AST and ALP were determined with enzymatic colorimetric assays using an autoanalyzer for clinical chemistry (Cobas 6000/c501; Roche Diagnostics GmbH, Rotkreuz, Switzerland).

### 2.5. Statistical Analysis

The statistical analysis of the plasma and serum metabolite data involved two parts. First, metabolomics datasets were analyzed using several modules, including the statistical, quantitative enrichment, pathway enrichment and biomarker analysis modules, of the web-based open access platform MetaboAnalyst 5.0 [14]. Second, differential analysis of the metabolomics and clinical biochemistry parameters was performed using SAS (version 9.4, SAS Inst. Inc., Cary, NC, USA). As the different feeding of the suckling piglets only started from day 10 of life, effects of the feeding were only analyzed for the time points thereafter. In each of the modules of MetaboAnalyst 5.0, the sample data were normalized to adjust for systematic differences among samples. In doing so, samples were normalized by sum and log transformed. After normalization the data were subjected to chemometrics analysis, specifically sparse Partial Least Squares-Discriminant Analysis (sPLS-DA), within the statistical analysis module. The ten most discriminant metabolites for each component were identified and pairwise score plots were used to identify characteristic trends or groupings among days of life. Based on the sPLS-DA results, the obtained metabolites were plotted according to their importance in separating the days of life based on the variable importance in the projection (VIP) scores and visualized in loading plots. The VIP score is a weighted sum of squares of the sPLS-DA loadings, whereby scores suggest that the selected variable is significantly involved in the separation of groups [15]. Only the features for the first component are presented here. The metabolites were ranked by the absolute values of their loadings. In total, three sPLS-DA were conducted, one for the whole dataset, and one each for the data subsets of lipids and metabolites other than lipids. This was done due to the targeted metabolomics approach, featuring plasma lipids, whereas the metabolites other than lipids covered a broad range of metabolic pathways. Biomarker analysis was performed using MetaboAnalyst 5.0. Samples were filtered by determining the relative standard deviation (RSD = standard deviation/mean) and normalized as described for the statistical module. To identify important features, receiver operating characteristic (ROC) curve analysis based on partial least squares discriminant regression was performed as a classification and feature ranking method. Results for the 15 most important features and the abundance between the feeding groups on individual days of life were displayed as signature features. Afterwards, pathway enrichment analysis was performed using the respective module. Pathway analysis used the KEGG Pathway Database (www.genome.jp/kegg/pathway.html) for Homo sapiens as a reference pathway library because no respective database was available for Sus scrofa. Previously, Lefort et al. [16] manually cross-checked the Homo sapiens database to confirm the relevance of the human pathways for the pigs.

Prior to the differential analysis in SAS, the residuals of the data for serum biochemistry and metabolome were tested for normal distribution using the Shapiro–Wilk test. If residuals were not normally distributed, they were converted using the Boxcox method and the Transreg procedure in SAS, before data were subjected to ANOVA using the MIXED procedure using repeated and random models. The first model for repeated measurements were used to assess whether differences in blood data existed over time. Fixed effects in this model included day of life, dietary treatment, sex and replicate batch and their two- and three-way-interactions. The experimental unit was piglet nested within litter. For many parameters, differences for sex and dietary treatment were not observed and not included in the final model for the respective parameter. Degrees of freedom were approximated by the Kenward–Roger method. Multiple pairwise comparisons among least-square means were performed using the probability of difference option in SAS. Data were expressed as least squares means ± standard error of the mean (SEM). Significance was defined at *p* ≤ 0.05 and trends were discussed at 0.05 < *p* ≤ 0.10.

## 3. Results

The RP-HPLC-MS/MS and IC-HR-MS analyses identified a total of 345 unique metabolites in plasma samples across all sampling time points and ages of piglets, of which about two-thirds represented metabolites from phospholipid, sphingomyelin and triglyceride metabolism. The majority of the metabolites could be detected in plasma at all age stages. Nevertheless, some triglycerides were not present in the plasma of piglets after weaning that received creep feed during the suckling period.

In order to categorize the metabolites with the highest discriminant power between days of life, we performed three separate sPLS-DAs. As the lipid fraction dominated the most discriminant metabolites, besides analyzing the whole data set, we analyzed the datasets for plasma lipids and water-soluble metabolites separately in order to distinguish important metabolites from the various pathways. Irrespective of piglets’ nutrition during the suckling phase, orthogonal projections of the sPLS-DA showed distinguishable metabolite profiles for the different days of life (Figure 2), with the greatest discrimination in plasma profiles between metabolomes before and after weaning.

The sources of variation for the ten most influential metabolites were displayed based on their VIP scores (VIP > 1), and are presented separately for the whole dataset and the two subsets of metabolites (Figure 3a–c). The VIP for the whole metabolite dataset showed that seven plasma triglycerides, hippuric acid and one phosphatidylcholine with diacyl residue sum C32:0 (PC aa C32:0) were the most influential metabolites (Figure 3a). Triglycerides predominated as the most discriminant lipids in the respective dataset (Figure 3b). For the water-soluble compounds, hippuric acid was the most influential metabolite (Figure 3c). Other major influential water-soluble metabolites were alpha- and beta-amino acids, including alpha-aminobutyric acid (AABA), taurine, proline, tyrosine, ornithine, serine, beta-alanine and arginine (Figure 3c).

As the next step, biomarker analysis using multivariate ROC curve based exploratory analysis with PLS-DA as classification method was applied to identify marker metabolites for the two nutritional groups for day 14, 21, 28, 31 and 35 of life. Signature feature ranking identified mainly plasma triglycerides as being characteristically different between the two nutritional groups from day 14 to 35 of life (Figure 4a–e), whereas water-soluble metabolites were less represented among the 15 marker metabolites at each day of life. Although triglycerides (TG) were among the top metabolites, the actual TG differed between the two feeding groups on the various days of life. Accordingly, plasma levels of TG 20:4_36:5, TG 17:0_34:2, TG 18:2_38:6 on days 14, 21 and 28 of life, respectively, were indicative for and higher in creep-fed piglets compared to piglets only receiving sow milk (Figure 4a–c). Postweaning, marker analysis showed that TG C17:1_C34:2 and TG C18:0_C32:1 on day 31 and 35 of life, respectively, were indicative for piglets that received sow milk only (Figure 4d,e). From the water-soluble metabolites, it is noteworthy that serotonin was among the top ten metabolites on day 28 of life and two bile acids on day 35 of life; all three metabolites were lower in creep fed piglets compared to piglets that were fed only sow milk during the suckling piglets.

Enrichment pathway analysis was conducted for water-soluble metabolites in plasma only; first to identify enriched metabolite sets and pathways (Figure 5, Figure 6 and Appendix A). The major pathways that were enriched on day 7 of life and declined towards day 14 of life were purine metabolism, propanoate metabolism, tricarboxylic acid cycle (TCA) and pyrimidine metabolism (Figure 5a). Compared to the suckling period, metabolites belonging to pathways within histidine metabolism, nitrogen metabolism and D-glutamine and D-glutamate metabolism were higher during the postweaning phase (Figure 5b). The most affected pathways by creep feeding were similarly but separately investigated for the suckling and postweaning period. During the suckling period, plasma samples of creep-fed piglets distinguished by higher concentrations of metabolites within pathways related to pyruvate metabolism, lysine degradation and tryptophan metabolism (Figure 6a). Postweaning, plasma of creep-fed piglets was enriched in metabolites related to pyrimidine metabolism, D-glutamine and D-glutamate metabolism and nitrogen metabolism (Figure 6b). Influential metabolites from the three top enriched pathways are provided in Appendix A.

Results for the differential analysis of the plasma metabolome using mixed models can be found in Appendix A. The influential plasma metabolites that were extracted by the sPLS-DA and pathway enrichment analysis were also different in the mixed models. For the majority of water-soluble metabolites, their plasma concentrations were mainly affected by age of the piglet, increasing or decreasing depending on the metabolite (Appendix A). For instance, differential analysis showed that primary and secondary bile acids increased in plasma from day 7 to day 28 of life and dropped postweaning. Likewise, the amino acid metabolome showed elevated levels of many proteinogenic amino acids as well as of taurine, citrulline and ornithine on day 7 and/or day 14 of life which declined afterwards. Concentrations of almost all lipids measured in plasma changed with age (*p* < 0.05; Appendix A). Specifically, the concentrations of TG in plasma dropped after weaning; many of them were below the detection limit postweaning, whereas effects of the feeding during the suckling period were small. Acyl-carnitines were mostly higher in creep-fed piglets from day 14 of life compared to piglets that only received sow milk. Creep feeding in the suckling phase lowered several phosphatidylcholines, sphingomyelins and one lysophosphatidylcholine on day 35 of life compared to piglets that only received sow milk.

The measured serum parameters of the clinical biochemistry were all affected by age (*p* < 0.05; Table 1). Serum concentration of total protein remained stable throughout the suckling phase but decreased postweaning (*p* = 0.016). The effect of feeding in the suckling period indicated that creep-fed piglets had lower serum total protein from day 28 of life and especially postweaning (*p* = 0.004). Serum levels of ALP decreased from day 7 to day 31 of life and remained at this level on day 35 of life. Serum ALT and AST mainly declined from day 7 to 14 of life and remained at this level (*p* < 0.05). Serum electrolytes (i.e., calcium, phosphate, sodium, potassium and chloride) were stable during the suckling phase and mainly dropped on day 31 of life to recover on day 35 of life (*p* < 0.05). Creep-fed piglets had lower serum calcium (*p* = 0.003) and sodium (*p* = 0.020) on day 31 of life compared to piglets that received only sow milk.

Body weight development of the piglets showed similar growth between the two feeding groups during the suckling phase, whereas on day 34 of life piglets that received creep feed during the suckling phase weighed 1.8 kg less than the piglets that only suckled sow milk during the suckling phase (*p* = 0.005; Figure 7).

## 4. Discussion

Normal development in the early neonatal period is essential for life-long performance and health in pigs [17,18]. Piglet’s development is driven by intrinsic (i.e., genetical programming) and extrinsic factors (e.g., nutrients) throughout the early life phases. Here, we provide missing data for postnatal concentrations of metabolites and metabolic markers in plasma and serum from day 7 of life until after weaning, which represent useful reference values for the practice to assess the nutritional, physiological and health status of very young piglets. In terms of the interpretation of the data, it should be noted that the presented plasma metabolome values are the net sum from intestinal absorption and hepatic and peripheral metabolism; hence, the summation from anabolic and catabolic reactions. In using targeted metabolomics, we had a strong focus on lipids and specific metabolic pathways related to sugar/energy, bile acid and amino acid/nitrogen metabolism, whereas other physiological pathways may have been underrepresented. Moreover, as no reference pathway library was available for *Sus scrofa*, the human KEGG pathway database was used to predict metabolome pathway enrichments. Although the similarity between the human and porcine genome amounts to about 90%, and despite the fact that the relevance of the human pathways for pigs was confirmed previously [16] (Lefort et al., 2020), differences in the annotations may exist, which is a weakness when applying the pathway enrichment analysis for species other than humans and has to be kept in mind when interpreting the present results.

The orthogonal projections of sPLS-DA provided the first idea that distinguishable metabolite profiles for the suckling and postweaning period could be obtained for the various days of life. Although metabolomes of the consecutive days in the suckling period overlapped, a certain trend for diverging plasma metabolomes was discernible, which was confirmed by the differential analysis and by protruding the VIP scores from the sPLS-DA models for the various days of life. The present age-related profiles showed evidence that the piglets’ metabolism altered within the first two weeks of life and pinpointed markers for the developmental stage of the suckling and newly weaned piglets. The sPLS-DA supported the importance of lipids in the metabolism of the neonatal piglets, identifying several TGs as the most influential metabolites with elevated concentrations in the early suckling phase, probably being indicative for the role of fat to meet their energy demand for thermogenesis and ATP generation. In addition, the identified lipids (i.e., TGs) and phosphatidylcholine [phosphatitylcholine diacyl (PC aa) C32:0] are important building blocks for membranes, hormones and body insulation. The three most influential, TG 18:0_30:0, TG 14:0_34:1 and TG 16:0_34:0 according to the VIP scores, may be useful marker metabolites for the suckling phase that largely drop postweaning. Part of the identified lipids were probably endogenously synthesized in hepatocytes. Nevertheless, it needs to be considered that the milk fatty acid profile and thus the nutrition of the gestating and lactating sow probably influenced the detected fatty acid profiles in the piglet plasma [19,20]. Therefore, the discriminant nature of the identified lipids from this experiment needs to be verified in further studies. Weaning largely affected the lipid profile. Many TG, acyl-carnitines and phospholipids dropped in their concentrations postweaning, often stronger on day 35 than on day 31 of life; probably being indicative for the lower feed and fat intake in the first week postweaning. Biomarker identification showed that TG 17:1_34:2, TG 17:0_36:1 and TG 18:0_32:1 were distinctive for the postweaning days and may be marker candidates for this period. However, similar to the identified lipids for the suckling phase, there is the dependence of the plasma lipid profile from the dietary lipid intake. Consequently, their validity needs to be proven in a further set of piglets being fed a different prestarter diet.

Our targeted approach revealed strong age-related concentration patterns of water-soluble metabolites in the first two weeks of life which matched the high anabolic rate, energy demand for thermogenesis and tissue growth in the piglets. Indeed, piglets from modern hybrid lines, like the present ones, double their body weight within the first two weeks of life. From the identified enrichment pathways, metabolites of the TCA (i.e., pyruvic acid, fumaric acid, succinic acid, aconitic acid and citric acid) and purine metabolism (i.e., urea and uric acid) as well as 2-hydroxybutyric acid, beta-alanine and histamine, were discriminated as key metabolites, being raised in plasma on day 7 compared to day 14 of life. The elevated levels of many proteinogenic amino acids together with the urea cycle metabolites arginine, citrulline and ornithine on day 14 of life may point towards a higher muscle protein turnover and deamination and thus higher utilization of amino acids as energy substrates. By contrast, concentrations of primary and secondary bile acids were indicative for the later suckling period. Although bile acid concentrations were probably the result of the higher milk and consequently milk fat intake on days 21 and 28 of life, they were also a sign for digestive maturation. Weaning appeared not to impact bile acid concentrations and composition because they were still high on day 31 of life. The question arises whether the drop in bile acid concentrations on day 35 of life was a delayed weaning effect due to the lower feed intake and/or the shift in diet composition with less fat and mainly plant-based lipids. Plasma glycine and taurine concentrations needed for the conjugation of primary bile acids were lower on day 35 of life, supporting a lower delivery of nutrients. In addition, the secondary bile acids (e.g., hyodeoxycholic acid) originate from microbial metabolism of bile salts [21] and were thus representative of microbial activity in the piglets’ gut, supporting high microbial activity on day 28 of life before weaning. Many bacterial taxa along the small and large intestines are capable of modifying the conjugated primary bile acids [22]. Therefore, it is difficult to pinpoint the drop in secondary bile acids to the bile salt hydrolase and/or bile acid-inducible enzyme activity of a specific taxon and gut site. Microbial modification of bile salts is essential for bile acid homeostasis [22] and their lower levels postweaning may have implications for mucosal functioning and inflammatory signaling [23]. Likewise, plasma acetate was rising from day 7 to 28 of life, suggesting increased intestinal fermentation of milk glycans [24] with the increasing milk intake during the suckling phase. Contrary to the observation for bile acids, weaning seemed not to impair intestinal fermentation as acetate concentration in plasma continued to rise postweaning. Propionate and butyrate were only detectable in plasma after weaning, demonstrating the importance of substrate (plant-based diet) for their intestinal production and subsequent systemic appearance. Results from differential analysis and of the VIP scores from the sPLS-DA models identified benzoic acid and its glycine-conjugate hippuric acid as discriminant metabolites postweaning. Since plasma benzoic acid and hippuric acid originate from the intake of plant polyphenols [25], their plasma concentrations may be useful markers for piglets’ plant intake postweaning. Indeed, their values would support increasing feed intake from day 31 to 35 of life. By contrast, largely raised plasma concentrations of phenylacetylglycine and alpha-aminobutyric acid on day 31 of life were indicative of catabolic reactions (i.e., phospholipid catabolism and methionine and threonine catabolism, respectively) [26] and hence body tissue mobilization. The enrichment pathway analysis further emphasized glutamate and glutamine as key metabolites within the most influential pathways related to amino acid turnover, which were elevated in plasma postweaning compared to the suckling period. Concurrently, urea cycle metabolites dropped postweaning. On the one hand, this could mean that deamination pathways were not upregulated for use of amino acids as energetic substrate postweaning. This is contrasted with the elevated concentration of alpha-aminobutyric acid on day 31 of life, which indicated body tissue mobilization. Therefore, our observation of lower urea cycle metabolites may be a sign of low availability of the precursors, i.e., arginine and ornithine, due to lower feed intake to form sufficient citrulline and urea [26], leading to the accumulation of glutamate and glutamine as key regulatory elements [27].

In modern pig breeding, weaning takes place earlier than naturally, shifting the piglet back to a glucogenic metabolism due to the carbohydrate-rich weaner diet. Creep feeding may have the same effect due to its high carbohydrate content. Therefore, we also characterized the impact of nutrition during the suckling phase as cause for variation in plasma metabolite concentrations, as this is relevant information for using the plasma metabolome data in practice. The commercial creep feed that we used in this study contained 29.4% starch and 26.8% sugar on dry matter basis, whereas the fat amounted to only 7.8% on a dry matter basis. Although the actual amount consumed by the individual piglets probably varied, creep feeding altered the fat and carbohydrate metabolism as indicated by the different development of plasma acyl-carnitines in creep-fed and sow milk-only-fed piglets. Acyl-carnitines are key factors regulating the balance of intracellular sugar and lipid metabolism [28]. Specifically, the plasma levels of free and acetyl-carnitine increased with creep feeding; suggesting that the higher rate of glycolysis due to the higher starch/glucose intake in the creep-fed piglets produced more acetyl-CoA which was then transported in the blood compared to piglets that only received sow milk. The enrichment pathway analysis further supported this assumption. Plasma concentrations of metabolites belonging to the pyruvate and glycolysis-gluconeogenesis pathways were elevated in the suckling period due to creep feeding. Likewise, results from differential analysis and signature features implied major alterations in plasma lipid fractions due to creep feeding. This was supported by the biomarker analysis which confirmed that the plasma lipid composition, especially TG, could distinguish which piglets received creep feed during the suckling period, e.g., TG 20:4_36:5, TG 17:0_34:2 and TG 18:2_38:6 on days 14, 21 and 28 of life, respectively. The identified TG were distinctively elevated by the creep feeding compared to only sow milk nutrition. Moreover, the differential analysis supported that creep-fed piglets may have used less fat as fuel, favoring glucose as an energy substrate, as most TG were elevated from day 14 to 28 of life in creep-fed piglets. Although plasma glucose levels were not different, the postprandial insulin response in the piglets was presumably higher after ingesting the starch-rich creep feed and prestarter diet, promoting hepatic lipid synthesis and rising blood triglyceride levels [29]. The creep feeding effect was not only detectable during the suckling phase, when the creep feed was provided, but lasted until after weaning; again major changes were noticeable in the postweaning TG profile. The ketone levels, i.e., 2-hydroxybutyrate and 2-keto isovaleric acid, were higher postweaning, particularly on day 31 of life, but not different between the feeding groups, supposing similar body fat mobilization due to low feed intake irrespective of the preweaning nutrition. However, the higher body weight on day 35 of life indicated that piglets that suckled only sow milk may have recovered faster from weaning stress and started eating more feed than piglets from the creep feed group. This is an interesting finding which rejects our hypothesis that the piglets from the creep feed group should have had an advantage by being adapted to consume dry feed and to the prestarter diet before weaning. Differences in feed intake may explain the diverging plasma lipid levels of creep-fed and only sow milk-fed piglets postweaning, becoming depleted in certain TG in the creep-fed piglets that were still detected in the plasma of only sow milk-fed piglets. Assumingly, the depletion may be related to a higher body fat mobilization in creep-fed piglets compared to sow milk-fed piglets.

In terms of clinical biochemistry parameters that are regularly used in routine diagnostics, developmental patterns were discernible for serum total protein, liver enzyme activities and electrolytes, demonstrating again the importance for specific reference values for neonatal piglets. The serum enzymes ALP, AST and ALT showed especially strong developmental patterns with high activities on day 7 of life which then declined to day 14. This observation likely reflected the high metabolic rate of the hepatocytes at this age, thereby supporting the findings for the plasma metabolome profile at this age, and not liver dysfunction. Plasma total protein and electrolytes were relatively stable during the suckling phase, whereas weaning and creep feeding were more influential, with postweaning serum levels being generally lower. Interestingly, creep-fed piglets showed a much stronger postweaning drop in serum total protein, ALP, calcium and sodium especially on day 31 of life, which, from a clinical perspective, may be interpreted as malnutrition [30] and may be additional useful markers for piglets feed intake postweaning.

In conclusion, the present plasma metabolome and serum biochemistry data provide conclusive information for the postnatal evolution of lipids and water-soluble metabolites and metabolic markers, which can be used as reference values for the practice to assess the nutritional, physiological and health status of neonatal piglets. Plasma lipid profiles in particular corresponded to the type of nutrition in the suckling phase and showed a strong weaning effect. Potential marker metabolites for age, creep feeding and weaning were identified. Influential TG 20:4_36:5, TG 17:0_34:2 and TG 18:2_38:6 were higher in creep-fed piglets on days 14, 21 and 28 of life, respectively, compared to only sow milk-fed piglets. Metabolites belonging to pathways within histidine, D-glutamine and D-glutamate metabolism as well as hippuric acid were distinctive for the postweaning compared to the suckling period. Selected lipids should be verified in future work due to their dependency of the dietary fatty acid composition.

## Figures and Tables

**Figure 1 metabolites-13-00214-f001:**
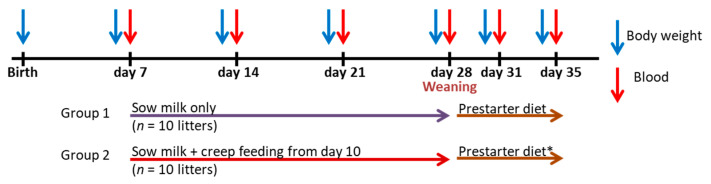
Outline of the experimental period. Ten litters only had access to sow milk during the suckling period (group 1 = sow milk group), whereas the other ten litters received additional creep feed from day 10 of life (group 2 = creep feed group). * On days 24 and 25 of life the creep feed was gradually mixed with the prestarter diet after which the piglets from the creep feed group were fed only the prestarter diet.

**Figure 2 metabolites-13-00214-f002:**
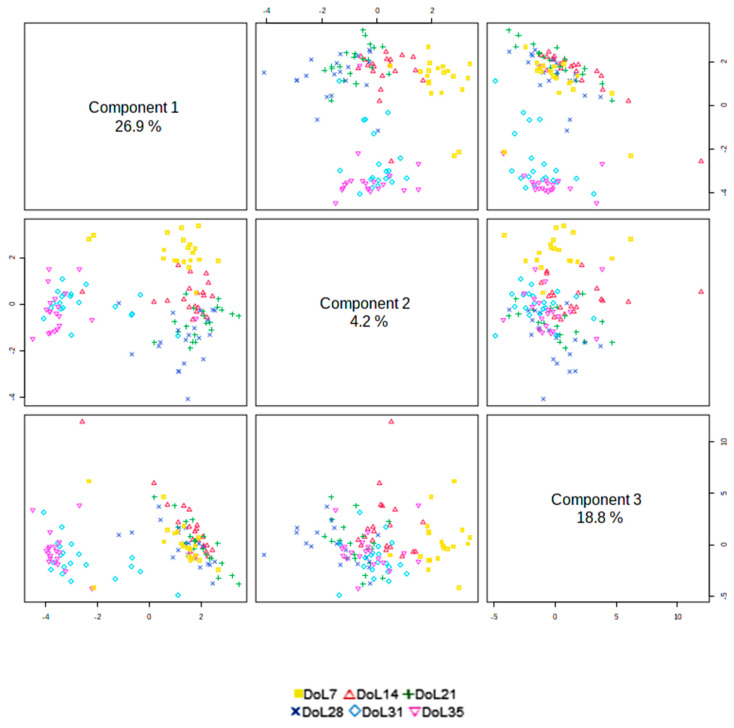
Orthogonal projections of the sparse partial least squares-discriminant analysis for component 1, 2 and 3 of plasma metabolomes in piglets for the different days of life [Day of life (DoL) 7, 14, 21, 28, 31 and 35; n = 20/DoL].

**Figure 3 metabolites-13-00214-f003:**
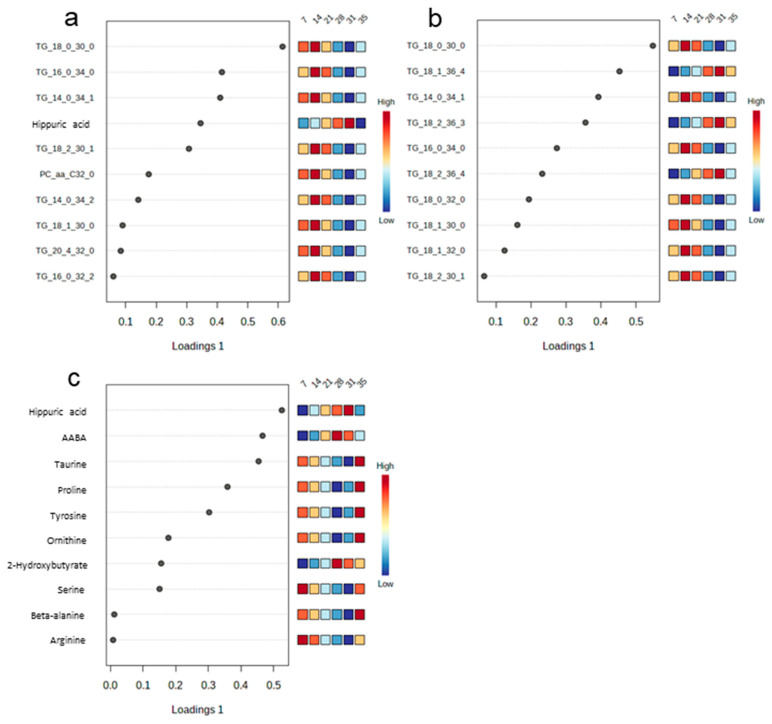
Visualization of results from partial least squares-discriminant analysis in loading plots according to their importance for the individual days of life based on the variable importance in the projection (VIP) scores. Most discriminant plasma metabolites in piglets among (**a**) all metabolites; (**b**) lipids; and (**c**) water-soluble metabolites. Piglets were weaned on day 28 of life. AABA, alpha-aminobutyric acid; PC aa, phosphatidylcholine diacyl; TG, triglyceride.

**Figure 4 metabolites-13-00214-f004:**
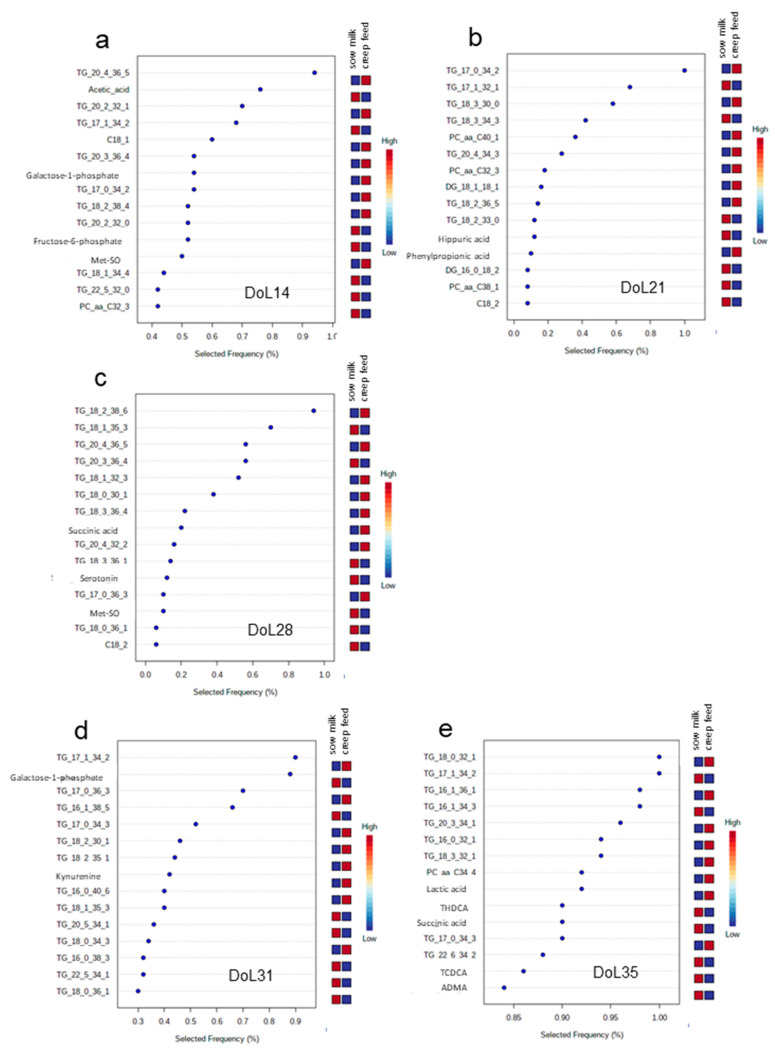
Biomarker analysis using multivariate receiver operating characteristic curve-based exploratory analysis with PLS-DA as classification method was applied to identify marker plasma metabolites for the two nutritional groups for the different days of life. Creep feed was fed from day 10 of life until weaning (day 28 of life). Most discriminant metabolites for day (**a**) 14; (**b**) 21; (**c**) 28; (**d**) 31; and (**e**) 35 of life. ADMA, asymmetric dimethylarginine; Met-SO, methionine sulfoxide; THDCA, taurohyodeoxycholic acid; TCDCA, taurochenodeoxycholic acid; TG, triglyceride.

**Figure 5 metabolites-13-00214-f005:**
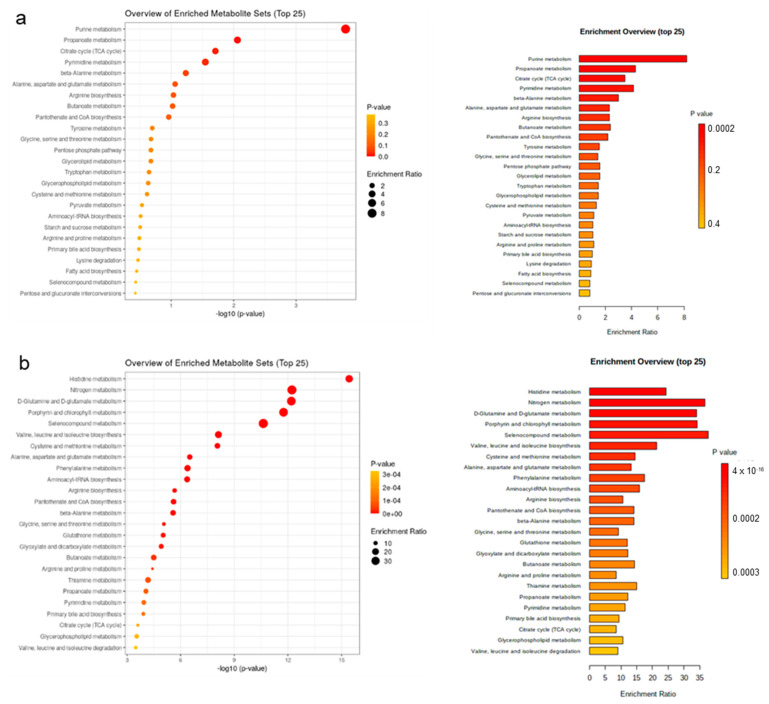
Enriched pathways (top 25) in discriminant plasma metabolites for sparse partial least squares-discriminant analysis. Pathway analysis used the KEGG Pathway Database for *Homo sapiens* as a reference pathway library because no respective database was available for *Sus scrofa*. Most discriminant pathways for (**a**) day 7 versus day 14 of life; and (**b**) suckling versus postweaning period. Piglets were weaned on day 28 of life.

**Figure 6 metabolites-13-00214-f006:**
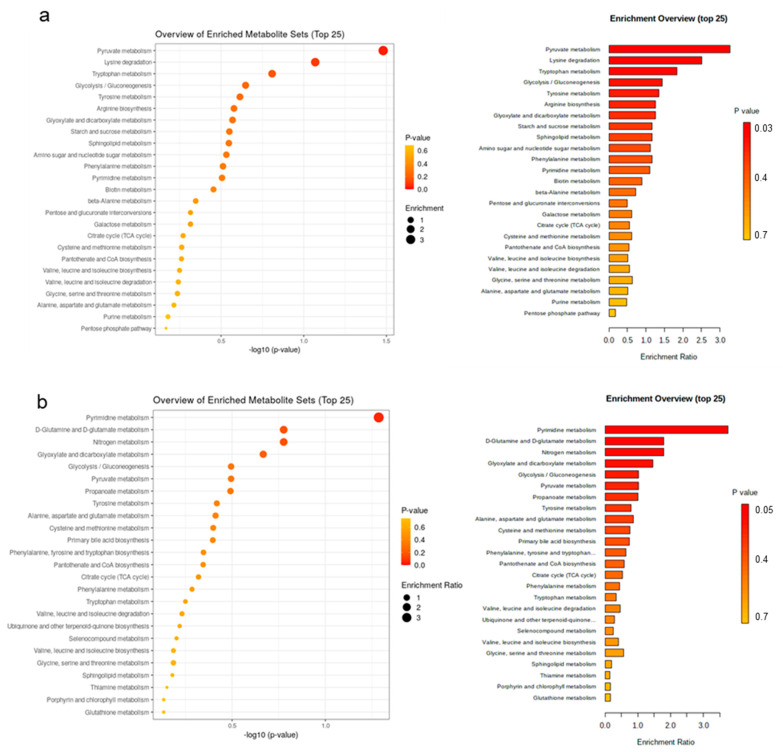
Enriched pathways (top 25) in discriminant metabolites for sparse partial least squares-discriminant analysis. Pathway analysis used the KEGG Pathway Database for *Homo sapiens* as a reference pathway library because no respective database was available for *Sus scrofa*. Most discriminant pathways for (**a**) creep feeding effect during the suckling period (only sow milk versus additional creep feeding); and (**b**) creep feeding effect during the postweaning period (only sow milk versus additional creep feeding in suckling period). Creep feed was fed from day 10 of life until weaning (day 28 of life).

**Figure 7 metabolites-13-00214-f007:**
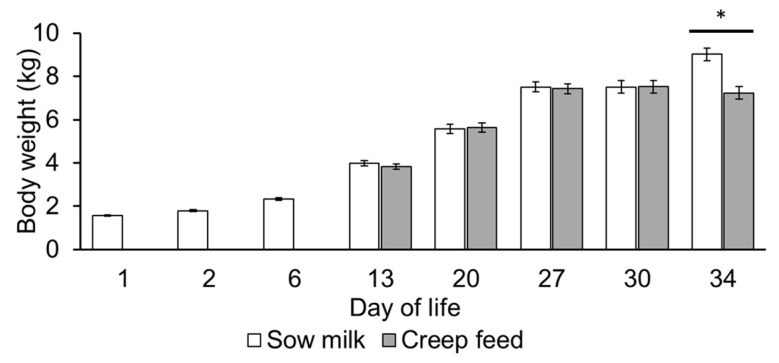
Body weight development of piglets during the suckling and early postweaning period. Piglets were weaned on day 28 of life (n = 5/feeding group and sex). * *p* < 0.05.

**Table 1 metabolites-13-00214-t001:** Differences in age-related development of clinical biochemistry parameters in serum of suckling and newly weaned piglets receiving only sow milk or additionally creep feed from day of life 10 *.

Day of Life (DoL)	7	14		21		28		31		35			*p* Value		
Feeding (Feed)	Sow Milk	Sow Milk	Creep Feed	Sow Milk	Creep Feed	Sow Milk	Creep Feed	Sow Milk	Creep Feed	Sow Milk	Creep Feed	SEM	DoL	Feed	DoL × Feed
Total protein (g/dL)	4.64	4.45	4.45	4.48	4.24	4.48	4.06	4.32	3.70	4.48	3.79	0.19	0.016	0.004	0.225
ALP (U/L)	1582	1084	1155	836	756	504	434	312	254	277	224	182.1	<0.001	0.653	0.998
AST (U/L)	80	42	35	34	36	57	36	31	27	35	34	9.18	<0.001	0.701	0.447
ALT (U/L)	54	31	29	31	28	35	35	43	46	36	38	4.78	<0.001	0.668	0.887
Calcium (mmol/L)	2.81	2.78	2.77	2.67	2.65	2.69	2.53	2.28	2.08	2.46	2.29	0.06	<0.001	0.003	0.482
Phosphate (mmol/L)	2.94	2.94	3.07	2.81	2.80	2.75	2.73	2.25	2.03	2.18	2.23	0.09	<0.001	0.996	0.540
Sodium (mmol/L)	137	136	135	137	134	136	133	130	125	135	132	1.88	<0.001	0.020	0.848
Potassium (mmol/L)	5.67	5.06	5.23	4.74	4.81	4.94	4.95	4.42	4.39	5.00	5.07	0.25	<0.001	0.483	0.987
Chloride (mmol/L)	99	99	98	100	97	99	97	96	93	99	98	1.56	0.023	0.082	0.821

* Values are least squares means ± standard error of the mean (SEM). Piglets were weaned on day 28 of life. ALP, alkaline phosphatase; ALT, alanine aminotransferase; AST, aspartate aminotransferase.

## Data Availability

Data is not publicly available due to privacy or ethical restrictions.

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
