# Peer review of "Creep Feeding and Weaning Influence the Postnatal Evolution of the Plasma Metabolome in Neonatal Piglets"

_metabolites, 2023, doi:10.3390/metabo13020214_

Round 1
Reviewer 1 Report
Comments for authors:
Title:
Creep Feeding and Weaning Influence the Postnatal Evolution of the Plasma Metabolome in Neonatal Piglets
Authors:
Metzler-Zebeli B.U., Lerch F., Yosi F., Vötterl J., Koger S., Aigensberger M., Rennhofer P.M., Berthiller F., Schwartz-Zimmermann H.E.
Manuscript ID: metabolites-2185227
Objective:
In the present article, the authors evaluated blood metabolites and metabolic markers in piglets fed with sow milk versus sow milk and offered creep feed from day 10 until weaning. Blood collection was performed on days 7, 14, 21, 28, 31 and 35. Plasma triglycerides were discriminative for age and nutrition within the suckling phase. Metabolite pathways for histidine, D-glutamine, D-glutamate metabolism and hippuric acid were distinctive in the case of the post-weaning phase.
Points of criticism:
The manuscript is very extensive, of high scientific quality and should be readable even by non-specialist readers. Numerous triglycerides are abbreviated, e.g., line 29 "20:4_36:5". For the general readership, it would be helpful to explain these abbreviations (annotation of TG including the abbreviations ae and aa).
The authors used the human pathway database (KEGG) because a corresponding Sus scrofa Domestica data source is unavailable. Genetically, there is a 90% match, but one must always refer to the differences, e.g., in the study's weakness – which has been completely lacking up to now. In addition, the food and husbandry of the mother animals are, of course, decisive factors influencing the data collected, which should also be stated in this respect.
The illustrations should be self-explanatory. Unfortunately, the illustrations are hardly legible - even when using the zoom setting. In this respect, the illustrations need to be improved, e.g., in Figure 2, at least on the upper right side, marking the "days".
The labelling of the figures in Figure 3 is correct up to Figure 3c, after which d and e should follow (as noted in line 307). This must be amended.
In Figure 3a, the “TG” under "Galactose 1 Phosphate" is not readable on the ordinate.
On page 12, the figure is labelled Figure 5. Figure 5 is already on page 11, so on page 12, it must be Figure 6. This must be changed.
Comment: Figure 6 shows that the piglets which suckled only sow milk had a significantly higher BMI on day 34. This should be discussed, particularly with the creep feed, which has the potential for improvement.
Due to the abundance of data and its complex analysis and statistical evaluation, the focus on the essentials sometimes needs to be recovered. Therefore, the authors should include a flow chart for data generation at the beginning and better highlight the "highlights" in conclusion.
In reference 10, the year is mentioned twice. This must be revised.
Author Response
Objective:
In the present article, the authors evaluated blood metabolites and metabolic markers in piglets fed with sow milk versus sow milk and offered creep feed from day 10 until weaning. Blood collection was performed on days 7, 14, 21, 28, 31 and 35. Plasma triglycerides were discriminative for age and nutrition within the suckling phase. Metabolite pathways for histidine, D-glutamine, D-glutamate metabolism and hippuric acid were distinctive in the case of the post-weaning phase.
AUTHORS: Thank you very much for your valuable comments. In considering your comments, we hope to have improved our manuscript.
Points of criticism:
The manuscript is very extensive, of high scientific quality and should be readable even by non-specialist readers. Numerous triglycerides are abbreviated, e.g., line 29 "20:4_36:5". For the general readership, it would be helpful to explain these abbreviations (annotation of TG including the abbreviations ae and aa).
AUTHORS: The abbreviations were explained in the text, table footnotes and figure legends.
The authors used the human pathway database (KEGG) because a corresponding Sus scrofa Domestica data source is unavailable. Genetically, there is a 90% match, but one must always refer to the differences, e.g., in the study's weakness – which has been completely lacking up to now. In addition, the food and husbandry of the mother animals are, of course, decisive factors influencing the data collected, which should also be stated in this respect.
AUTHORS: We added a statement in the Discussion section to discuss the weakness of using the human pathway database (Lines 426-433). The mother sows were housed and fed similarly. Indeed, the mother sows are major factors influencing the outcome of the study. We added a statement in the Materials and Methods section about the mother sows (Lines 92-98).
The illustrations should be self-explanatory. Unfortunately, the illustrations are hardly legible - even when using the zoom setting. In this respect, the illustrations need to be improved, e.g., in Figure 2, at least on the upper right side, marking the "days".
AUTHORS: Thank you. We tried to improve the quality and legibility of the legend of the ordinates. The graphs were generated in MetaboAnalyst. The software allows limited modification of the legends of the x- and y-ordinates.
The labelling of the figures in Figure 3 is correct up to Figure 3c, after which d and e should follow (as noted in line 307). This must be amended.
AUTHORS: Corrected.
In Figure 3a, the “TG” under "Galactose 1 Phosphate" is not readable on the ordinate.
AUTHORS: Corrected. We tried to improve the readability of the parameters on the ordinates.
On page 12, the figure is labelled Figure 5. Figure 5 is already on page 11, so on page 12, it must be Figure 6. This must be changed.
AUTHORS: Thank you. We corrected this error (new Figure 7).
Comment: Figure 6 shows that the piglets which suckled only sow milk had a significantly higher BMI on day 34. This should be discussed, particularly with the creep feed, which has the potential for improvement.
AUTHORS: Thank you for this suggestion. A statement was added to the Discussion section to discuss this finding (Lines 550-559).
Due to the abundance of data and its complex analysis and statistical evaluation, the focus on the essentials sometimes needs to be recovered. Therefore, the authors should include a flow chart for data generation at the beginning and better highlight the "highlights" in conclusion.
AUTHORS: A flow chart was provided in the Materials and Methods sections (new Figure 1). Several ‘highlights’ were added in the Conclusion.
In reference 10, the year is mentioned twice. This must be revised.
AUTHORS: corrected.
Reviewer 2 Report
Creep feeding is a management practice that presents opportunities for improving weaning weights and post-weaning pig performance. Therefore, the aim of the study should be considered important from the point of view of piglet rearing efficiency. The work analyzes changes in plasma metabolome and serum biochemistry in suckling and newly weaned piglets
The layout of the work and the content of individual chapters do not raise any objections. The title is in fact a research hypothesis The abstract was properly prepared, as was the introduction. The goal was correctly formulated. The methodology has been developed in detail and transparently.
In the abstract (line 27-28)it is written: “Blood was collected from one piglet/litter on days 7, 14, 21, 28, 31 and 35 of life, totalling to five females and five males/group/day”. There is no detailed description in the methodology. This needs supplementing. Admittedly, in the "results" section, it is stated (line 391): „n = 5/feeding group and sex”, but it is difficult to say unequivocally based on the methodology where this number comes from.
The work is based on a very extensive results and supplementary materials. Discussion based on the correct interpretation of the obtained results, the conclusion was correctly formulated. The work does not raise any other methodological and substantive objections. After the indicated minor addition, it can be published in the Metabolites Journal.
Author Response
Creep feeding is a management practice that presents opportunities for improving weaning weights and post-weaning pig performance. Therefore, the aim of the study should be considered important from the point of view of piglet rearing efficiency. The work analyzes changes in plasma metabolome and serum biochemistry in suckling and newly weaned piglets.
The layout of the work and the content of individual chapters do not raise any objections. The title is in fact a research hypothesis The abstract was properly prepared, as was the introduction. The goal was correctly formulated. The methodology has been developed in detail and transparently.
In the abstract (line 27-28)it is written: “Blood was collected from one piglet/litter on days 7, 14, 21, 28, 31 and 35 of life, totalling to five females and five males/group/day”. There is no detailed description in the methodology. This needs supplementing. Admittedly, in the "results" section, it is stated (line 391): „n = 5/feeding group and sex”, but it is difficult to say unequivocally based on the methodology where this number comes from.
The work is based on a very extensive results and supplementary materials. Discussion based on the correct interpretation of the obtained results, the conclusion was correctly formulated. The work does not raise any other methodological and substantive objections. After the indicated minor addition, it can be published in the Metabolites Journal.
AUTHORS: Thank you very much for your comments. We were pleased to read that there were not major issues. We improved the description of the blood collection and tried to clarify the piglet numbers in the Materials and Methods section (Lines 141-150).